# Estimating the Peak Outflow and Maximum Erosion Rate during the Breach of Embankment Dam

**Mahmoud T. Ghonim** [1], **Ashraf Jatwary** [1], **Magdy H. Mowafy** [1], **Martina Zelenakova** [2,*], **Hany F. Abd-Elhamid** [1,2], **H. Omara** [3] **and Hazem M. Eldeeb** [1]

1    Department of Water and Water Structures Engineering, Faculty of Engineering, Zagazig University, Zagazig 44519, Egypt; mtghonim@gmail.com (M.T.G.); ashrafhakem@zu.edu.eg (A.J.); magdy_mowafy@yahoo.com (M.H.M.); hany_farhat2003@yahoo.com or hany.abdelhamid@tuke.sk (H.F.A.-E.); hmeldeeb@eng.zu.edu.eg (H.M.E.)
2    Department of Environmental Engineering, Faculty of Civil Engineering, Technical University of Kosice, 040 01 Kosice, Slovakia
3    Irrigation and Hydraulics Department, Faculty of Engineering, Tanta University, Tanta 31527, Egypt; hewida@f-eng.tanta.edu.eg or hewida.omara@ejust.edu.eg
*    Correspondence: martina.zelenakova@tuke.sk; Tel.: +421-905-985-765

**Abstract:** Understanding and modeling a dam breaching process is an essential investigation, because it aims to minimize the flood's hazards, and its impact on people and structures, using suitable mitigation plans. In the current study, three-dimensional numerical modeling is carried out using the FLOW-3D HYDRO program to investigate the impact of various factors, including the dam grain size materials, crest width, inflow discharge, and tail water depth on the dam breach process, particularly the peak outflow, and the erosion rate. The results show that changing the grain size of the dam material from fine sand to medium and coarse sand leads to an increase in the peak outflow discharge by 16.0% and the maximum erosion rate by 20.0%. Furthermore, increasing the dam crest width by 40% leads to a decrease in the peak outflow by 3.0% and the maximum erosion rates by 4.50%. Moreover, increasing the inflow discharge by 25.0% increases the peak outflow by 23.0% and the maximum erosion rates by 21.0%. Finally, increasing the tail water depth by 50.0% leads to decreasing the peak outflow by 4.50% and the maximum erosion rate by 43.0%. The study findings are considered of high importance for dam design and operation control. Moreover, the results can be applied for the optimum determination of the crest width and tail water depth that leads to improving the dam stability.

**Keywords:** dam breach; numerical modeling; overtopping failure; computational fluid dynamics (CFD); flood risks

## 1. Introduction

Embankment structures, such as dams and levees, are classified according to the purpose, size, capacity, and height of the structure. Moreover, they may be non-erodible, erodible, or mixed-type. Dams are usually constructed for flood control, producing electricity, and providing water. The safety and stability of these structures depend on many factors, including the construction materials and configurations. There are several causes of embankment failures, which can be categorized as piping (35%), overtopping (25%), spillway erosion (14%), excessive deformation (11%), sliding (10%), gate failure (2%), defect construction (2%), and earthquake instability (2%) [1]. It is noticeable that the percentage of overtopping failures could be considereable high. Therefore, this study focuses on investigating dam breach due the overtopping. Overtopping is the main reason for flood events, which have become a common natural disaster and which present significant challenges. A number of studies propose an integrated approach based on remote sensing data and simplified hydrodynamic models to assess flood hazards [2].

Dam breaching is a complicated event that is affected by several factors. Seepage, overtopping, sliding, and different geotechnical parameters of dam material could all affect the dam breach process. The breaching development of landslip dams is significantly impacted by erosion caused by the overtopping flow. However, the dam breaching studies that are now in use do not adequately account for these two processes. At the water–soil interface, the deposition of soil slows down the flow of eroded particles and lowers the water velocity. The outburst flood flow increases and the breaching duration lowers if soil deposition is ignored, as it is in many of the current models [3].

The probability distribution of embankment dam breach parameters including height, side slope, width, and time of formation, were investigated using a correlation analysis technique depending on the dam failures database (3861 observations) [4]. Yang et al. [5] carried out various experimental cases to investigate the effect of the angle of repose for soil grains on the dam breaching process. Coleman et al. [6] constructed homogenous uniform non-cohesive small embankments to analyze the effect of overtopping flows under a constant water level condition. It was found that the breach channel eroded the downstream face of the embankment and rotated about a fixed pivot point. The location of the pivot point is presented as a function of the gain size of embankment material.

The effect of the dam geometry parameters, such as dam slopes and crest width, on the breaching of non-cohesive homogenous embankment dams was examined [7]. It was concluded that the dam geometry parameters have a noticeable effect on dam breaching at stage1 (rapid erosion) and do not have any significant effect during the second stage (gradual evolution toward the equilibrium state). Zhu et al. and Jiang [8,9] constructed experimental models of a non-cohesive homogenous embankment dam to investigate the impact of modifications to the downstream slope angle, dam height, and flume gradient on the dam breaching process. Furthermore, the eroded particles from the dam body accumulate in two main zones at the middle and lower parts of the downstream dam face. Furthermore, increasing the flume gradient or downstream slope angle accelerates the dam breach process. Based on the experimental program to investigate the effect of changing the downstream slope angle, dam crest width, and dam composition material on the dam breaching evolution [10], it was concluded that, reducing the downstream slope angle and widening the dam crest led to decreased rates of erosion. On the other hand, increasing the non-uniformity coefficient for the dam materials led to an increase in the rates of erosion initially, which then decreased at the final stages.

Wu et al. [11] performed a variety of cohesive and non-cohesive embankment dam studies in the lab. They concluded that dam breaching was divided into four stages, namely slope erosion, longitudinal gully, lateral erosion, and relative stabilization. Moreover, it was noticed that the lateral expansion in non-cohesive embankment dams is non-symmetrical by coefficients approximately from 2.20 to 2.60. Rifai et al. [12] caried out 54 experiments to examine the influences of inflow discharge, grain size, and cohesion of the dam material on the non-cohesive homogenous embankment dams. They concluded that the inflow discharge has a noticeable effect on the dam breach evolution, while the grain size (fine material where apparent cohesion) has a slight impact on the dam breach.

Dynamic 3D photogrammetry is a good technique in experimental work to track breach evolution in detail due to rapid changes in both flow characteristics and dam geometry [13]. Yusof et al. [14] presented an experimental model and numerical simulation by FLOW-3D for investigating the behavior of the dam breaching process. Moreover, it was found that the width of the breach is narrower at the middle part of the embankment than at the upstream and downstream faces. Also, a relationship between the amount of volume loss from the dam body to the hydrograph produced from the breach was presented.

Shen et al. [15] performed a numerical model for dam breaching by overtopping based on the dam morphological characteristics and experimental models. The results showed that the peak outflow, breach width, and time of peak discharge are more sensitive to the erosion mode.

The depth-averaged shallow water equation (SWE) has been frequently used to simulate the sediment–water mixture flow over recent decades, so many hydraulic engineering problems can be simulated successfully [16–19]. Wu et al. [16] presented a double-layer and two-phase flow SWE depending on non-cohesive embankment overtopping cases to simulate the dam-break flow over erodible sediment beds. Using the double-layer averaged model to simulate non-cohesive embankment dams minimizes the numerical instability compared with the single-layer models [20]. Onda et al. [21] used 3D to simulate the dam breaching process under the effect of both seepage and overtopping flows. From previous studies, it was concluded that the dam geometry has a noticeable effect on dam breaching and 3D models are more precise in simulating the dam breaching process and are fully coupled with sediment transport equations. The results of 3D models are more accurate than all previous 2D numerical studies, because the breach formation is affected by the third direction. Therefore, this study focuses on investigating the dam crest width, grain size, inflow discharge, and tail water depth effects on the peak outflow and erosion rates by using 3D models. Each parameter under investigation leads to a significant impact on the erosion process. For example, increasing the inflow discharge leads to an increase in the erosion rate and therefore accelerates dam failure.

To apply the current study, the FLOW-3D HYDRO software is a good for presenting the overtopping process in 3D, which depends on the finite volume method (FVM). The study findings may be considered of high importance for dam design and operation control, by determining the optimal dam crest width during the design stage and best tailwater depth during operation to achieve dam stability.

## 2. Numerical Modeling

The FLOW-3D HYDRO software is considered a powerful tool in the computational fluid dynamics (CFD) field, because it includes equations for the sediment movement and allows the construction of 3D models for complex hydraulic problems such as dam breaching processes [22]. FLOW-3D HYDRO is used in this study for simulating the dam breach process under different scenarios, such as grain size, crest width, inflow discharge, and tail water depth.

### 2.1. Governing Equations

The Reynolds–Averaged Navier–Stokes (RANS) equations are employed as the governing equations for the motion of the incompressible viscous fluid. The purpose of these equations is to pair with the volume and area porosity functions. Fractional Area/Volume Obstacle Representation approach (FAVOR) is the term applied to this formulation. This method is applied to the modeling of complex geometric areas for incompressible fluids [22].

The equation of continuity take the following form [23]:

$$\frac{\partial}{\partial x}(uA_x) + \frac{\partial}{\partial y}(vA_y) + \frac{\partial}{\partial z}(wA_z) = 0 \tag{1}$$

where, $(x, y, z)$ are the cartesian coordinates, $(u, v, w)$ are the velocity components in $(x, y, z)$ directions, and $(A_x, A_y, A_z)$ are the area fractions.

The momentum equations for the components of fluid velocity $(u, v, w)$ of the Navier–Stokes equations are in the three coordinate directions, with the following extra terms [23]:

$$\frac{\partial u}{\partial t} + \frac{1}{V_F}\left(uA_x\frac{\partial u}{\partial x} + vA_y\frac{\partial u}{\partial y} + +wA_z\frac{\partial u}{\partial z}\right) = \frac{-1}{\rho}\frac{\partial P}{\partial x} + G_x + f_x - b_x$$

$$\frac{\partial v}{\partial t} + \frac{1}{V_F}\left(uA_x\frac{\partial v}{\partial x} + vA_y\frac{\partial v}{\partial y} + +wA_z\frac{\partial v}{\partial z}\right) = \frac{-1}{\rho}\frac{\partial P}{\partial y} + G_y + f_y - b_y \tag{2}$$

$$\frac{\partial w}{\partial t} + \frac{1}{V_F}\left(uA_x\frac{\partial w}{\partial x} + vA_y\frac{\partial w}{\partial y} + +wA_z\frac{\partial w}{\partial z}\right) = \frac{-1}{\rho}\frac{\partial P}{\partial z} + G_z + f_z - b_z$$

where, $V_F$ is the fractional volume open to flow, $P$ is the average hydrodynamic pressure, $(G_x, G_y, G_z)$ are the body accelerations, $(f_x, f_y, f_z)$ are the viscous accelerations, and $(bx, by, bz)$ are flow losses across porous baffle plates. The volume of fluid (VOF) algorithm is employed to track the profile of the water surface at each time step. The Renormalized group (RNG) model achieves high accuracy compared with (K-$\omega$) and (K-$\varepsilon$) models [24]. Therefore, this model is employed to simulate the flow turbulence characteristics. Two main equations represent the RNG model, one for the turbulent kinetic energy $K_T$ and the other for its dissipation $\varepsilon_T$.

$$\frac{\partial k_T}{\partial t} + \frac{1}{V_F}\left(uA_x\frac{\partial k_T}{\partial x} + vA_y\frac{\partial k_T}{\partial y} + wA_z\frac{\partial k_T}{\partial z}\right) = P_T + G_T + Diff_{K_T} - \varepsilon_T \tag{3}$$

$$\frac{\partial \varepsilon_T}{\partial t} + \frac{1}{V_F}\left(uA_x\frac{\partial \varepsilon_T}{\partial x} + vA_y\frac{\partial \varepsilon_T}{\partial y} + wA_z\frac{\partial \varepsilon_T}{\partial z}\right) = \frac{C_1.\varepsilon_T}{K_T}(P_T + c_3.G_T) + Diff_\varepsilon - c_2\frac{\varepsilon_T^2}{k_T} \tag{4}$$

where, $K_T$ is the turbulent kinetic energy, $P_T$ is the turbulent kinetic energy production, $G_T$ is the buoyancy turbulence energy, $\varepsilon_T$ is the turbulent energy dissipation rate, $Diff_{K_T}$ and $Diff_\varepsilon$ are terms of diffusion, $c_1 = 1.44$, $c_2 = 1.92$ and $c_3 = 0.20$ are constant dimensionless parameters.

The sediment transport processes, which include bed load transport, suspended load transport, entrainment, and deposition, are simulated using the sediment scour model available in FLOW-3D HYDRO [22]. For the erosion process, Equation (5) illustrates how the standard wall function is used to predict the fluid shear stress on the bed surface [25].

$$k_{s,i} = C_{s,i} * d_{50} \tag{5}$$

where, $k_{s,i}$ is the Nikuradse roughness and $C_{s,i}$ is a user-defined coefficient.

The critical shear stress depends on a dimensionless parameter, the critical shields number, as shown in Equation (6) [26].

$$\theta_{cr,i} = \frac{\tau_{cr,i}}{\|g\|d_i\left(\rho_i - \rho_f\right)} \tag{6}$$

where, $\theta_{cr,i}$ is the critical shields number, $\tau_{cr,i}$ is the critical bed shear stress, $g$ is the absolute value of gravity acceleration, $d_i$ is the diameter of sediment grain, $\rho_i$ is the density of the sediment species ($i$), and $\rho_f$ is the density of the fluid.

The critical shields number is estimated using the Soulsby–Whitehouse equation [26].

$$\theta_{cr,i} = \frac{0.3}{1 + 1.2d_{*,i}} + 0.055[1 - exp(-0.02d_{*,i})] \tag{7}$$

where, $d_{*,i}$ is the dimensionless diameter of sediment and is presented at Equation (8), [26].

$$d_{*,i} = d_i\left[\frac{\rho_f(\rho_i - \rho_f)\|g\|}{\mu_f^2}\right]^{\frac{1}{3}} \tag{8}$$

where, $\mu_f$ is the fluid dynamic viscosity.

The discretized version of the Navier–Stokes system shows a linear relationship between pressure and velocity, and vice versa. Pressure–velocity coupling is the name given to this inter-equation interaction. It takes a unique approach to achieving pressure–velocity coupling. These techniques are SIMPLE, SIMPLER, SIMPLEC and PISO. The main technique applied in FLOW-3D is SIMPLE. According to this technique, an approximation of the velocity field is achieved by solving the momentum equation. The pressure distribution from the previous iteration, or a preliminary estimate, is used to compute the pressure gradient term. The new pressure distribution is obtained by formulating and solving

the pressure equation. After that, the correction of velocities is done, and a new values calculation of conservative is achieved. The implicit technique is employed to solve both momentum and pressure correction equations, while the explicit technique is applied to solve velocity correction [22].

Iterative numerical methods for solving fluid flow and heat transfer equations often use iteration procedures, requiring convergence criteria to determine when iterations can be terminated. Relaxation techniques, such as over-relaxation and under-relaxation, are often used in iteration methods to accelerate convergence and achieve numerically stable results in incompressible flow conditions. The choice of convergence criteria can significantly impact the quality of the results and computational time, requiring careful consideration of the appropriate amount of over- or under-relaxation. Computational fluid dynamics software users face challenges in selecting relaxation and convergence criteria, due to the varying problem specifics and the lack of universal guidelines. Despite standard recommended criteria, users often resort to trial-and-error adjustments for optimal results. FLOW-3D users can bypass these difficulties by selecting the relaxation and convergence criteria themselves, which are adjusted dynamically. However, they can override these criteria for special cases, such as large convergence criteria [22].

### 2.2. The Dam Geometry

In the current study, the dam dimensions; 0.20 m height ($H_o$), 0.10 m top width ($L_k$), 0.90 m bottom width ($W_o$), 0.20 m length ($B_o$), and U.S. and D.S. slopes ($2H : 1V$) are selected according to the most common previous experimental studies [27]. The spatial dam (i.e., a dam with an initial channel created through the dam's body) 3D geometry is prepared by SketchUp software and converted to stereolithography (STL) formatting, which is allowed in the FLOW-3D HYDRO program, as shown in Figure 1.

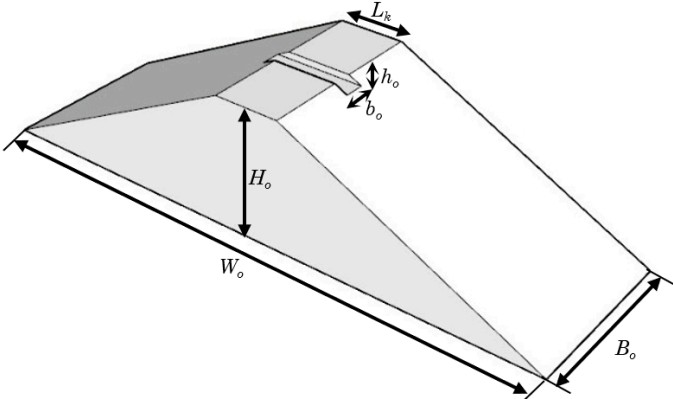

**Figure 1.** Dam geometry STL file.

### 2.3. Meshing of the Model

The meshing process in the FLOW-3D HYDRO program depends on simple rectangular orthogonal elements. Moreover, to minimize the time consumed by the numerical model, two mesh blocks were used, one coarse, and the other fine, as shown in Figure 2. Each mesh block consisted of a uniform cell of size 0.012 m for the coarse block and 0.006 for the fine block. These sizes were identified and selected after performing a gird convergence test. This test showed that the optimal grid size is 0.006 m for recording the dam profile evolution, because the value of RMSE became near to zero and a grid size smaller than 0.006 m leads to small variation in RMSE as shown in Figure 3.

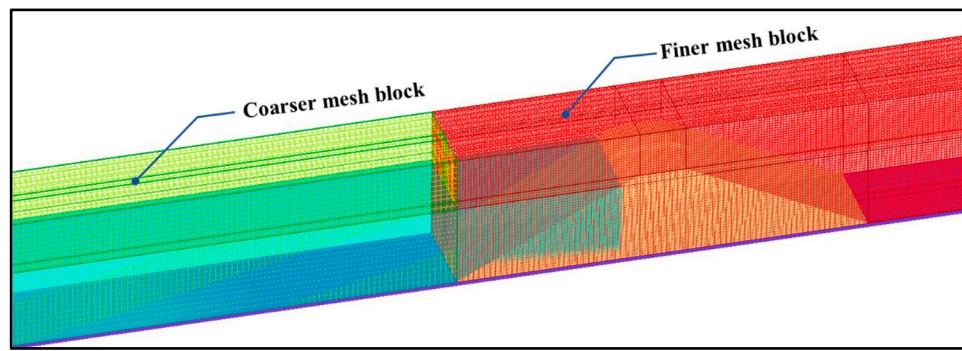

**Figure 2.** Mesh blocks.

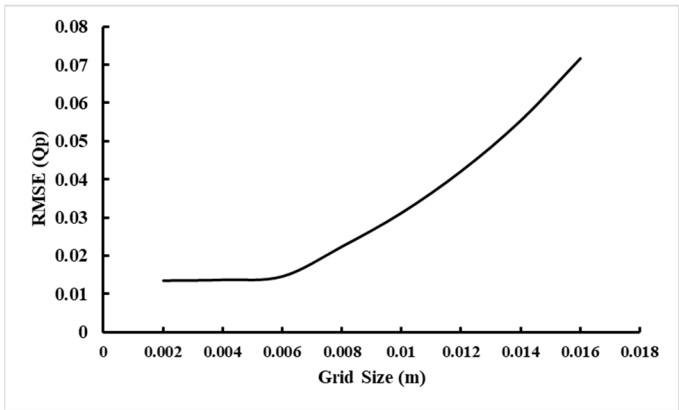

**Figure 3.** RMSE for different grid sizes.

The numerical model dimensions were identified in the three directions, as follows:

In the x-direction, the length is 3.56 m. The first mesh block consised of two mesh planes, which were fixed at distances of 0.00 m and 1.66 m with a uniform cell size of 0.012 m, and the second mesh block consisted of five mesh planes, which were adjusted at distances 1.66, 2.06, 2.16, 2.56, and 3.56 m with a uniform cell size of 0.006 m.

In the y-direction, the width is 0.20 m. The first and second mesh blocks consisted of four mesh planes, which were fixed at distances of 0.00 m and 0.20 m, while the other two other mesh planes were adjusted according to the initial breach width.

In the z-direction, the height is 0.30 m. The first and second mesh blocks consisted of four mesh planes, which were fixed at distances of 0.00, 0.20, and 0.30 m, while the remaining mesh plane was adjusted according to the initial breach depth, as shown in Figure 4.

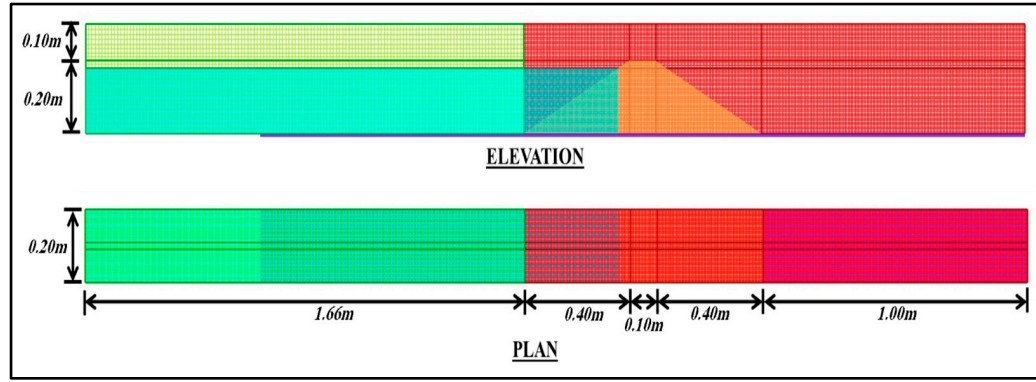

**Figure 4.** Mesh block dimensions as simulated by FLOW-3D HYDRO.

### 2.4. Boundary Conditions

The numerical model boundary conditions are considered an important factor, so the procedure of defining the boundary conditions needs to be done with caution, to avoid any errors in the simulation that increases the accuracy of the model. For the coarser mesh block, $X_{Min}$, $Y_{Min}$, and $Y_{Max}$ boundary conditions are specified like a wall, and the $X_{Max}$ is specified like symmetry. Moreover, the $Z_{Min}$ is specified like a volume flow rate, and the $Z_{Max}$ is specified like a pressure with value equal to the atmospheric pressure. Furthermore, for the finer mesh block, the $X_{Min}$, is defined as the symmetry boundary condition, the $X_{Max}$ is specified like outflow, the $Z_{Min}$, $Y_{Min}$, and $Y_{Max}$ are specified like a wall, and the $Z_{Max}$ is specified like pressure with atmospheric pressure, as shown in Figure 5.

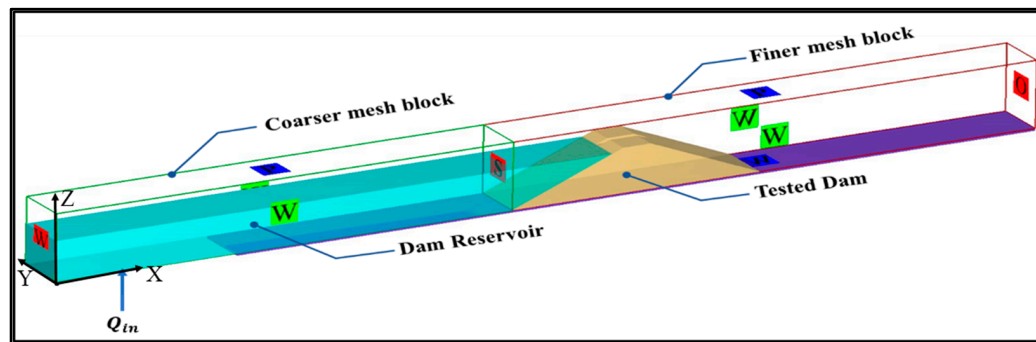

**Figure 5.** The boundary conditions of the numerical model.

### 2.5. Time Step Control

The Courant number, which manages the flow's distance throughout the simulation time step, is employed to determine the size of the maximum time step. In the simulation, the Courant number was adjusted to 0.25 in order to prevent the flow from going through more than one cell in a time step. The maximum time step value is 0.00075 s based on the Courant number value.

### 2.6. Numerical Model Validation

To achieve high model accuracy, the results were compared with a previous experimental program. Schmocker and Hager (2012) [27] presented an experimental program consisting of 31 tests by changing six variables (mean diameter $d_{50}$, initial dam height $H_o$, initial dam length $B_o$, crest width $L_K$, entrance flow distance $X_D$, and inflow discharge $Q_{in}$). A straight rectangular open glass-sided flume of 0.4 m width and 0.7m height was used for carrying out all the experimental tests. Test No. 1 from this experimental program was selected to confirm that the numerical model is accurate. Furthermore, throughout the validation stage, a few significant assumptions were made. It is assumed that the flow is three-dimensional, turbulent, viscous, and incompressible. Finally, the material of the dam is non-cohesive and homogeneous. The following variables were utilized in Test No. 1, as shown in Table 1:

**Table 1.** Variables of Test No.1.

| Variable | Value |
|---|---|
| Mean diameter ($d_{50}$) | 0.31 mm |
| Initial dam height ($H_o$) | 0.2 m |
| Initial dam length ($B_o$) | 0.2 m |
| Width of dam crest ($L_K$) | 0.1 m |
| Entrance flow distance ($X_D$) | 1.0 m |
| Inflow discharge ($Q_{in}$) | 6.0 lit/s |
| U.S and D.S dam slopes ($S_u$ & $S_d$) | 2H:1V |

**Table 1.** *Cont.*

| Variable | Value |
|---|---|
| Mass density ($\rho_s$) | 2650 kg/m$^3$ |
| Angle of repose ($\phi$) | 32° |
| Type of soil | Homogenous and non-cohesive |

The evolution of longitudinal dam profiles with time for both the numerical model and experimental work proves that the current numerical model achieves good matching with the experimental work, as shown in Figures 6 and 7.

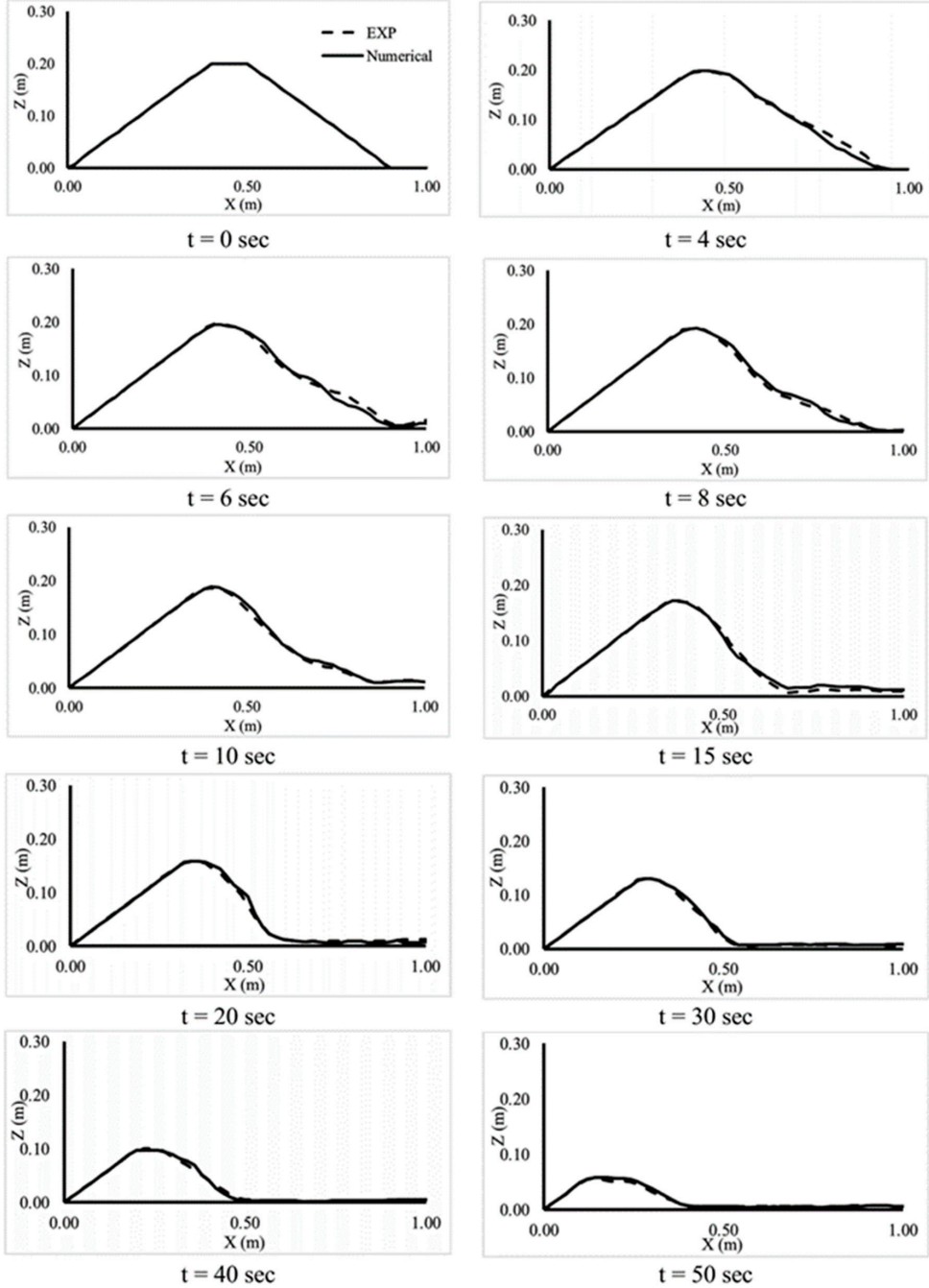

**Figure 6.** The evolution of longitudinal dam profiles with time for both the numerical model and experimental work.

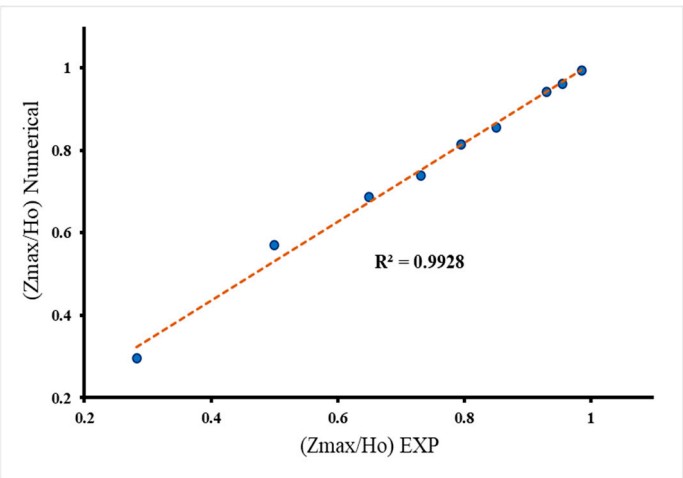

**Figure 7.** The experimental ratio of the maximum dam height at different times to the initial dam height versus numerical value.

## 3. Results and Discussion

The present study is investigates the impacts of various factors, including grain size materials, dam crest width, inflow discharge, and tail water depth on the peak outflow discharge ($Q_P$) and erosion rate (E) on the embankment dam. Depending on the different numerical parameter values identified at the validation stage, this work is divided into four scenarios. The first scenario considers the change of the dam grain size from fine sand to medium and coarse. The second scenario investigates the change of dam crest width. The third scenario studies the change in the inflow discharge value. The final scenario considers the change of the tail water depth.

These scenarios were performed under identical conditions, including reservoir water level, reservoir storage capacity, and dimensions of the checked dam, where the dam height ($H_o = 0.20$ m), dam length ($B_o = 0.20$ m), upstream and downstream dam slopes ($2H : 1V$), initial dimensions of breach (depth and width = 10% of the dam height), and homogenous or non-cohesive soil.

### 3.1. The Effect of Changing the Grain Size

To determine how the grain size affects the development of the dam breaching process, three cases were performed with the same value of the inflow discharge ($Q_{in}$ = 1.0 lit/s) and with a different grain size for each case. The first median grain diameter was 0.31mm "Fine sand", the second median grain diameter was 1.0 mm "Medium sand", and the third median grain diameter was 3.0 mm "Coarse sand" [28]. To make a comparison between different cases, the erosion rate, the ratio between maximum height of dam at different times ($Z_{Max}$) to the initial dam height ($H_o$), and also the breach outflow hydrograph, are calculated for each case. According to this scenario, the results indicate that increasing the dam grain size leads to increased erosion rates, because fine soil behaves as cohesive soil. The rates of erosion at the beginning rapidly increase until the maximum erosion rate occurs then decrease slowly, as shown in Figure 8. The maximum erosion rates mainly occur after the dam crest is totally eroded. The ratio between the maximum height of the dam at different times ($Z_{Max}$) to the initial dam height ($H_o$) is a good indicator for the erosion rate in the vertical direction. Figure 9 shows the ratio of maximum height at different times for different soil diameters. The results showed that the maximum elevation of the dam quickly decreases with increasing grain size. Furthermore, Figure 10 presents the relationship between the dam reservoir storage volume and time. This graph indicates that a significant amount of water has moved downstream since the maximum grain size had the lowest value of storage volume. This can be attributed to the maximum grain size having a smaller contact surface between the particles, which led to a decrease in soil resistance to the applied shear stress from water flow. On the other hand, this size

makes the rolling process for particles easier, which increases the instability during the overtopping process.

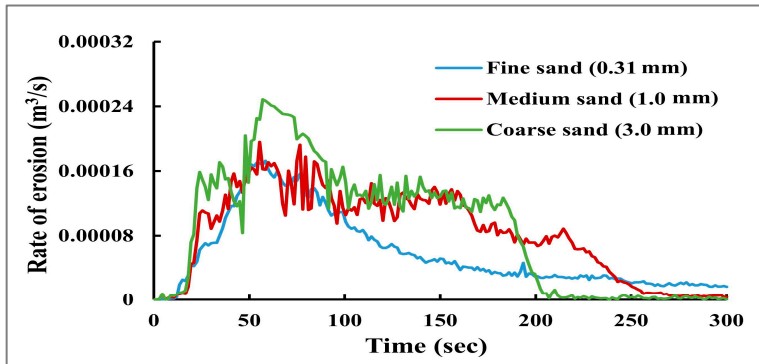

**Figure 8.** The rate of erosion for different soil diameters.

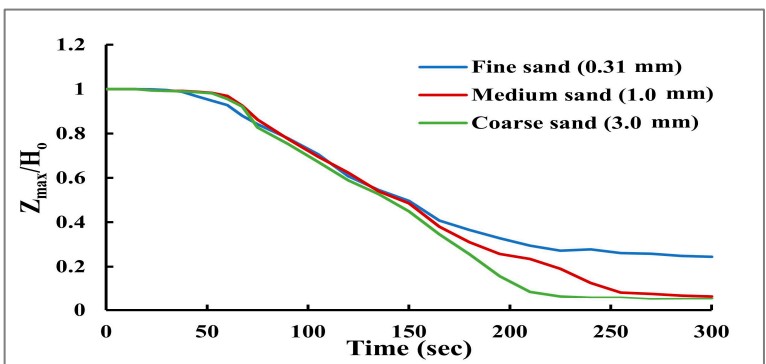

**Figure 9.** The ratio of maximum height at different times for different soil diameters.

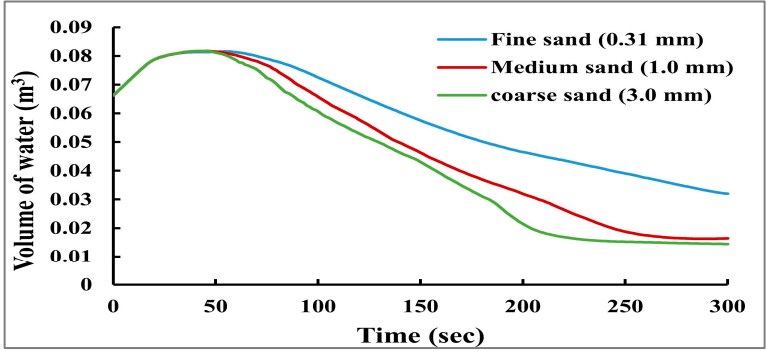

**Figure 10.** Dam storage volume for different soil diameters.

Figure 11 shows the breach outflow hydrograph. The peak outflow discharge and its time ($t_P$) are also studied for the three cases. The outflow discharge at the beginning increases rapidly until the peak outflow occurs and then decreases slowly until the value of the outflow is equal to the inflow discharge. The results show that increasing the dam grain size leads to decreasing the time of peak outflow discharge ($t_P$) and increasing the value of peak outflow discharge. There is a 16% variation in the peak outflow values across the three cases.

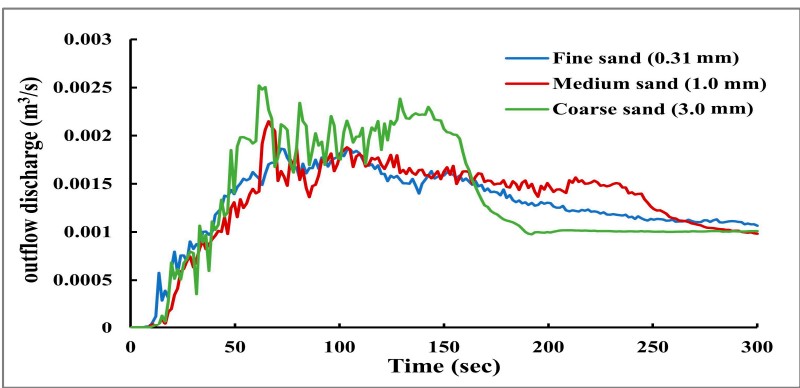

**Figure 11.** Breach outflow hydrograph for different soil diameters.

### 3.2. The Effect of Dam Crest Width

To examine the effect of the dam crest width on the evolution of dam breaching, three widths of crest (0.1, 0.15 and 0.2 m) are checked with the same value of the inflow discharge ($Q_{in}$ = 1.0 lit/s) to identify the worst-case scenario for the dam breaching. According to this, the findings indicate that the maximum erosion rate decreases by 4.50% when increasing the dam crest width by 40%. The maximum erosion rate occurs earlier in the case of small crest width than in a large width because of reaching the inflection point rapidly, as shown in Figure 12. The maximum erosion rate usually occurs after reaching the inflection point (i.e., the point connecting the U.S dam face with the dam crest) because the overtopping flow will increase due to the water retained in the reservoir. The top elevation of the dam decreases quickly with decreased dam crest width, as shown in Figure 13. The findings also illustrate that a significant amount of water has moved downstream since the minimum crest width had the lowest value for the storage volume, as shown in Figure 14. These findings show that, compared to the two other scenarios, a lowest crest width accelerates the process of dam breakdown and increases the rate of erosion. This effect could be related to the effect of the erosion direction, where at the beginning, the erosion occurs horizontally and parallel to the dam crest. After that, when the crest of the dam is totally eroded, the erosion direction becomes vertical, which leads to excessive erosion. In that case, a minimum crest width leads to a rapid transformation of erosion direction from horizontal to vertical and therefore accelerates dam failure.

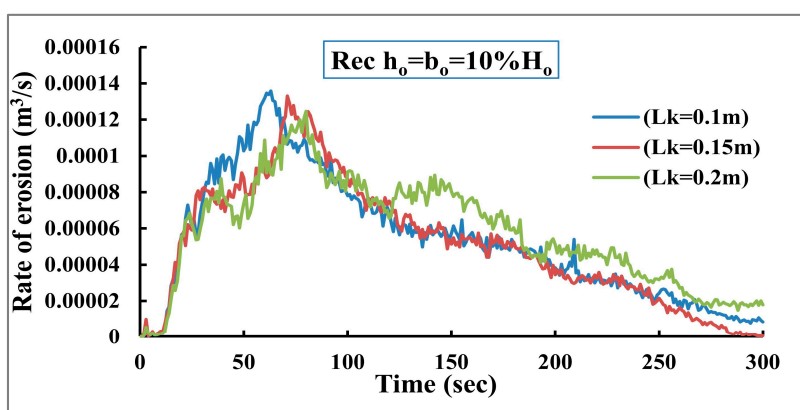

**Figure 12.** The erosion rate for different crest widths.

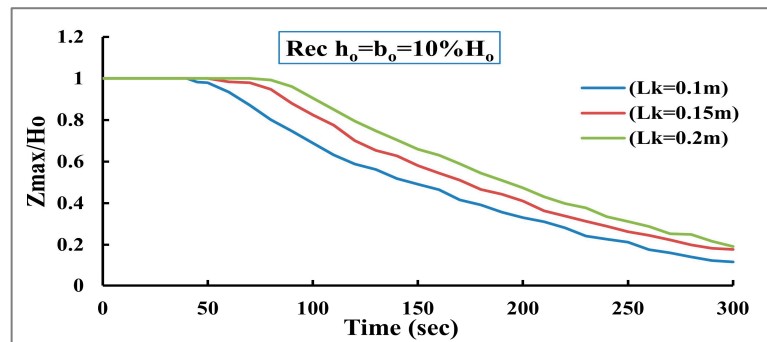

**Figure 13.** The maximum height with different times for different crest widths.

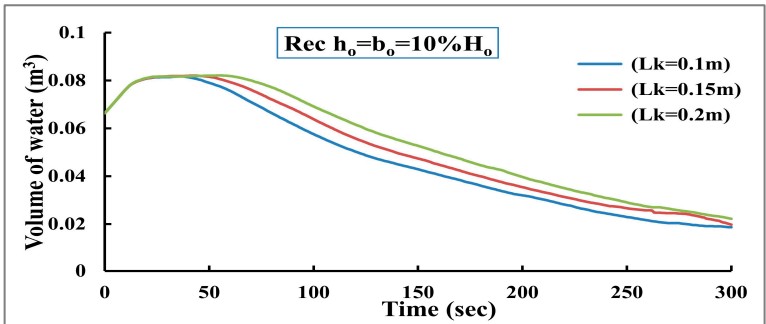

**Figure 14.** Dam storage volume rate for different crest widths.

Both the peak outflow time ($t_P$) and discharge are examined for the three cases. The results show that increasing the width of dam crest has increased the time of the peak outflow discharge and decreased the value of the peak outflow discharge. There is a 3.0% variation in the peak outflow values across the three cases, as shown in Figure 15.

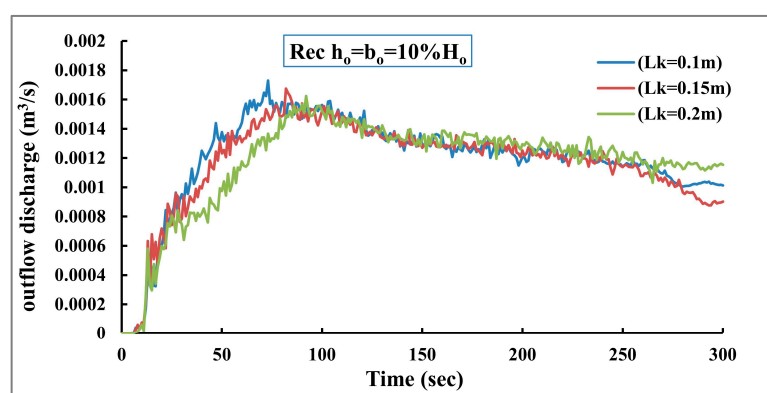

**Figure 15.** Breach outflow hydrograph for different crest widths.

### 3.3. The Effect of Inflow Discharge

To assess how the inflow discharge affects the development of the dam breaching, three inflow discharge values are tested to find the worst-case scenario for the dam breakdown. The first inflow value is 1.0 lit/s, the second is 1.25 lit/s, and the third is 1.57 lit/s. The results indicate that the maximum erosion rate increased by 21% with increasing the inflow discharge by 25%. The maximum erosion rate occurs earlier when increasing inflow discharge, as shown in Figure 16. Figure 17 shows the top elevation of the dam decreases quickly when increasing inflow discharge. Moreover, the results show that a significant amount of water has moved downstream since the maximum inflow discharge value had the lowest value for the storage volume as presented in Figure 18. In addition, compared to the other two scenarios, the maximum inflow discharge value causes an acceleration of the

dam break down process and an increase in the rate of erosion. The main reason for this behavior can be returned to increasing flow velocities, which leads to an increased applied shear stress and therefore accelerates dam failure.

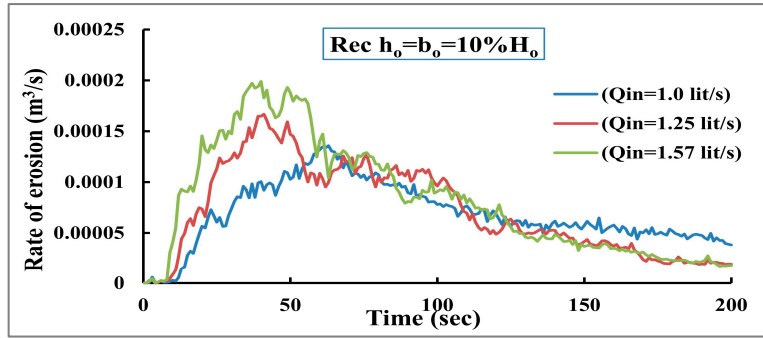

**Figure 16.** The rate of erosion for different inflow discharges.

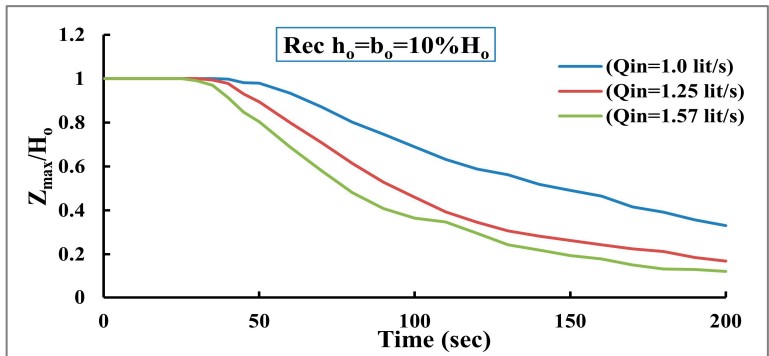

**Figure 17.** The maximum height versus time for different inflow discharges.

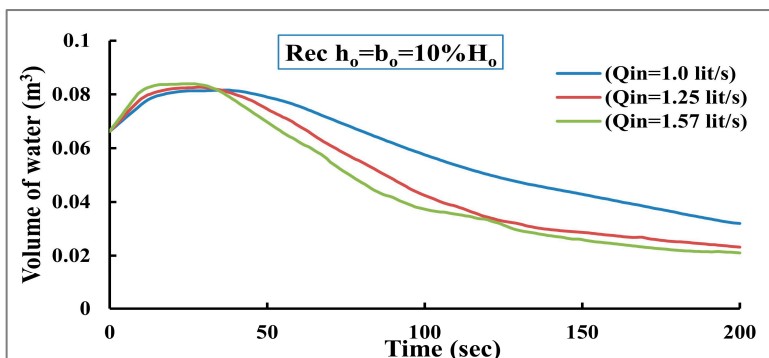

**Figure 18.** Dam storage volume for different inflow discharges.

Furthermore, the time of occurrence ($t_P$) and peak outflow discharge are investigated for the three cases. The findings indicate that increasing the inflow discharge decreased the time of peak outflow discharge and increased the peak outflow discharge. There is a 23% variation in the peak outflow values across the three cases, as shown in Figure 19.

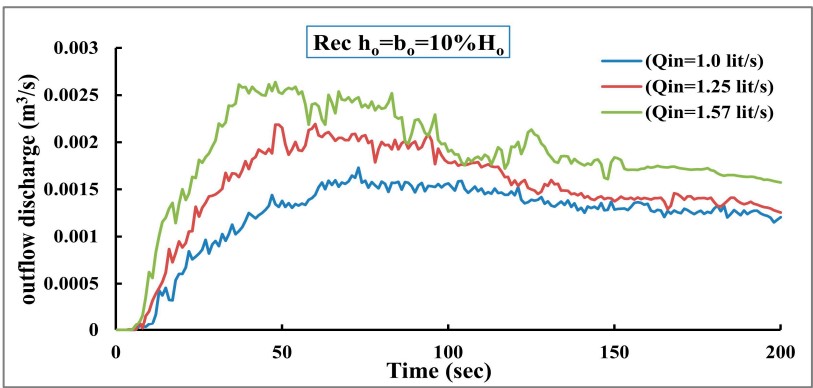

**Figure 19.** Breach outflow hydrograph for different inflow discharges.

### 3.4. The Effect of Tailwater Depth

Three depths are examined to identify the worst-case scenario for the dam failure to examine the impact of the tailwater depth (the depth of water at the downstream region) on the development of the dam breaching process. The first depth is 0.0 m (there is no tailwater depth), the second depth is 0.04 m, and the third depth is 0.06 m. The findings illustrate that the maximum erosion rate decreases by about 43% when increasing the tailwater depth by 50%. The maximum erosion rates are significantly affected by changing the tailwater depth, as shown in Figure 20, because any increase in the tailwater depth leads to increasing the stability of downstream dam slope and cause erosion to occur in small parts of the D.S dam slope. The difference between the erosion rates in the vertical direction is relatively small, as presented in Figure 21. Furthermore, the minimum tailwater depth has increased the erosion rate, which appears in the dam profile evolution and makes the dam break down rapidly compared with the two other cases, as shown in Figure 22.

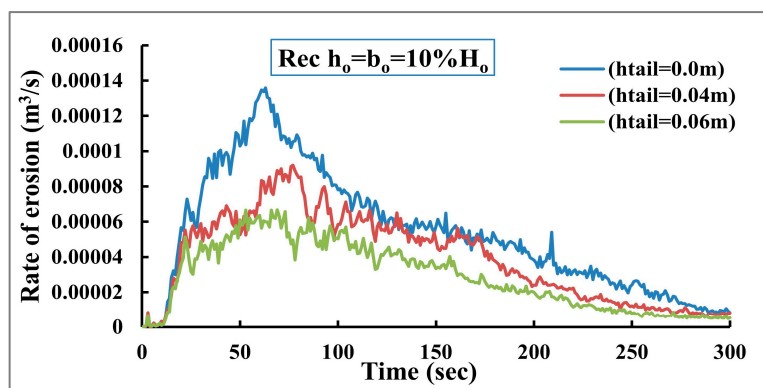

**Figure 20.** The erosion rate for different tail water depths.

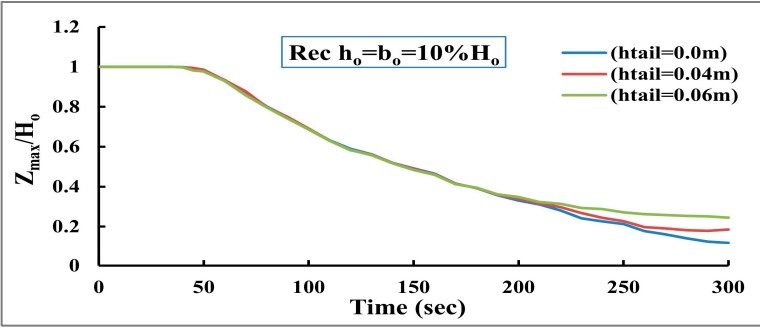

**Figure 21.** The ratio of maximum height with different times for different tail water depths.

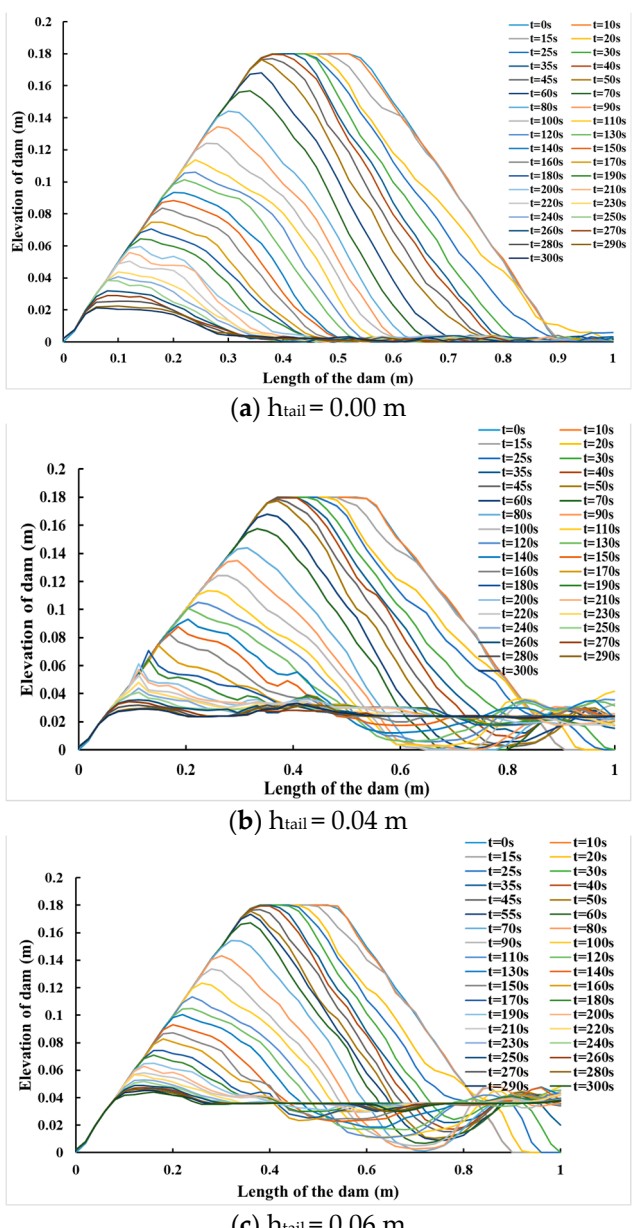

**(a)** h_tail = 0.00 m

**(b)** h_tail = 0.04 m

**(c)** h_tail = 0.06 m

**Figure 22.** Dam's Profile Evolution for different tailwater depths.

The peak outflow time ($t_P$) and discharge are also analyzed for the three cases. The results show that increasing the tail water depth causes the peak outflow discharge time ($t_P$) to increase and the peak outflow discharge value to decrease. There is a 4.50% variation in the peak outflow values across the three cases, as shown in Figure 23.

When equalization occurs between upstream water levels and downstream levels, it rapidly stops the dam failure process [29]. This confirms that increasing the tail water depth is a valuable method to increase the stability of the dam. Furthermore, using finer non-cohesive material in the dam body makes its behavior like cohesive soil and therefore increases dam stability against overtopping failure [30]. This also agrees with the current study, where a large grain size achieves instability for the dam during the overtopping process.

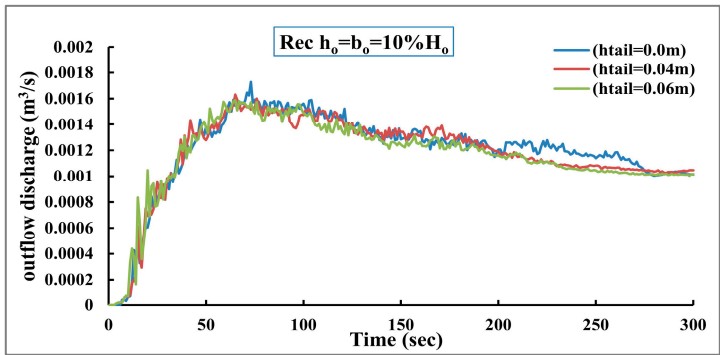

**Figure 23.** Breach outflow hydrograph for different tailwater depths.

## 4. Conclusions

The safety of embankment dams is an important issue to minimize flood hazards, so the dam breach simulation process should be performed with high accuracy. This study investigates the impact of different factors, such as: grain size diameter, dam crest width, inflow discharge and tail water depth on the peak outflow discharge ($Q_P$) and erosion rate (E) of embankment dam. The numerical modeling was performed using FLOW-3D HYDRO program. The results of this study show that increasing the dam grain size ($d_{50}$) increased both the maximum rate of erosion and the peak outflow by 20% and 16%, respectively. Furthermore, increasing dam crest width ($L_K$) by 40% decreased both the maximum rate of erosion and peak outflow discharge by 4.50% and 3.0%, respectively. Moreover, increasing the inflow discharge ($Q_{in}$) by 25% increased both the maximum rate of erosion and peak outflow by 21% and 23%, respectively. Finally, increasing the tailwater depth ($h_{tail}$) by 10% from the dam height decreased both the maximum rate of erosion and peak outflow by 43% and 4.50%, respectively. From previous findings, the tail water depth can be used as a good tool to manage the dam failure process by saving it with high levels. The study findings are considered of high importance for dam design and operation control. Moreover, the results can be applied for the optimum determination of crest width and tail water depth, which leads to improved dam stability. To increase the research information, there are some parameters that can be studied numerically, such as using a mix of cohesive and non-cohesive soils, wave action during dam overtopping, exploring the relationships between the parameters and the dam breach characteristics by using correlation analysis, performing regression analysis to develop mathematical models that relate the independent parameters to the peak outflow and erosion rates, and conducting a sensitivity analysis to assess the sensitivity of the dam breach process to variations in the parameters.

**Author Contributions:** All authors whose names appear on the submission made substantial contributions. Conceptualization, M.T.G., A.J., M.H.M., M.Z., H.F.A.-E., H.O. and H.M.E.; methodology, A.J., M.H.M., H.M.E., M.T.G. and H.F.A.-E.; data curation, A.J., M.T.G., H.O. and H.M.E.; validation, M.T.G., A.J., M.Z., H.F.A.-E. and H.O.; formal analysis, A.J., M.T.G., M.H.M., H.M.E. and H.F.A.-E. The first draft of the manuscript was written by A.J., M.H.M., M.T.G. and H.M.E.; revision and suggestions for the previous versions of the manuscript, H.M.E., M.T.G., A.J., M.H.M., M.Z., H.F.A.-E. and H.O. All authors have read and agreed to the published version of the manuscript.

**Funding:** This research received no external funding.

**Data Availability Statement:** Data are contained within the article.

**Acknowledgments:** This work was supported by the Slovak Research and Development Agency under contract no. APVV-20-0281. This article was created with the co-financing by the Governments of Czechia, Hungary, Poland and Slovakia through Visegrad+ Grants from International Visegrad Fund, project ID: 22220131.

**Conflicts of Interest:** The authors declare no conflict of interest.

**Abbreviations**

The following symbols are used in this paper:

| | |
|---|---|
| $A_x, A_y, A_z$ | Area fractions |
| $B_o$ | Initial dam length (m) |
| $b_o$ | Initial breach width (m) |
| $b_x, b_y, b_z$ | Flow losses in across porous baffle plates (m) |
| $d_{50}$ | Mean sediment diameter (m) |
| $E$ | Rate of erosion (m³/s) |
| $f_x, f_y, f_z$ | Viscous accelerations (m/s²) |
| $G_x, G_y, G_z$ | Body accelerations (m/s²) |
| $H_o$ | Initial dam height (m) |
| $h_o$ | Initial breach depth (m) |
| $h_{tail}$ | Tailwater depth (m) |
| $L_K$ | Dam crest width (m) |
| $P$ | Averaged hydrodynamic pressure (N/ m²) |
| $Q_{in}$ | inflow discharge (m³/s) |
| $Q_P$ | Peak outflow discharge (m³/s) |
| $t_P$ | Time of peak outflow (s) |
| $u, v, w$ | Velocities component in x-, y-, and z-direction (m/s) |
| $V_F$ | Fractional volume |
| $Z_{max}$ | Maximum dam height at different times (m) |
| $\rho$ | Density of the fluid (kg/m³) |

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
