# Peer review of "Estimating the Peak Outflow and Maximum Erosion Rate during the Breach of Embankment Dam"

_water, doi:10.3390/w16030399_

Round 1
Reviewer 1 Report
Comments and Suggestions for Authors
This paper helps to develop the field under investigation by simulating the breach of an embankment dam and examining various parameters related to real-world dams. However, deficiencies were observed during the review of this study, which, by fixing them, can potentially be published in a reliable scientific journal.
1. Most of the papers reviewed in the introduction are outdated, and the structure of the introduction should be improved. In the last three years, significant advancements in research related to breaches of embankment dams have occurred. To enhance the introduction, I recommend incorporating new papers as well as the following ideas:
· Background: Provide a brief overview of the significance of understanding and modeling the dam breaching process, emphasizing the importance of mitigating flood hazards and minimizing their impact on people and structures.
· Summary of previous findings: Provide a concise summary of the results obtained from previous studies.
· Research objective: Clearly state the objective of your study, which is to investigate the impact of different parameters on the dam breach process, specifically focusing on peak outflow and the rate of erosion.
· Methodology: Mention the specific numerical modeling approach you used to conduct the study. Briefly explain why this methodology suits the research objective and review previous simulations.
· Parameters under investigation: Highlight the parameters you examined in your study, including dam grain size materials, crest width, inflow discharge, and tailwater depth. Explain their relevance to the dam breach process and how they can influence peak outflow and erosion rates.
· Implications and applications: Discuss the significance of the study findings for dam design and operation control. Emphasize how the results can be applied to improve dam stability and optimize the determination of crest width and tailwater depth.
· Importance: Conclude the introduction by reiterating the high importance of the study findings and how they contribute to the existing knowledge in dam breaching. Highlight the potential impact of this research on minimizing flood hazards and improving mitigation plans.
2. The first sentence of subsection 2.1 is repeated in the previous paragraph. Please delete it.
3. Previous research shows several suitable software options for simulating the dam breaching process, including FLOW-3D HYDRO, HEC-RAS, MIKE 21, ICM, and HEC-HMS/HEC-FIA. What was the reason for using FLOW-3D HYDRO? Explain its strengths and advantages over other simulators.
4. Explain in full detail all assumptions and simplifications in the simulation process.
5. Why did you use the RNG k-e turbulence model? Have you evaluated the performance of other models, such as standard k-e and k-ω, through trial and error? If this is done, provide the results. Also, present the constants of the turbulence model in the relevant section.
6. The optimal grid significantly impacts the calculation cost and simulation accuracy. Present the results of the grid study in the form of a graph or the form of a table.
7. Show clearly the type of boundary condition in Figure 4. The current form is not legible
8. The paper is full of grammatical and typographical errors (acomplex in Line 121; thesimulation in Line 203). The paper needs substantial editing to improve its quality. It is recommended to seek a professional editor's help to refine the paper.
9. STL stands for what phrase? Mention the full form of abbreviations on first use.
10. All equations must be referenced. Additionally, modify any incorrect subscripts and superscripts within the manuscript. Sort the nomenclature table alphabetically.
11. What is the pressure-velocity coupling algorithm for the numerical solution? What techniques are used to discretize different equations? What are the convergence criteria for different equations? Add these items in full detail in the manuscript.
12. Simply presenting the results is inadequate; providing a thorough physical explanation for all observed alterations is essential. Expand the discussion on the results and organize a comparison with the results of other researchers to emphasize your findings. Since 2015, numerous papers close to this paper have emerged, offering opportunities for comparative analysis.
13. Suggestion to improve the paper: Explore the relationships between the parameters and the dam breach characteristics. Use correlation analysis to determine if there are any strong or weak correlations between the parameters and the peak outflow and erosion rates. This can help identify which parameters have the most influence on the dam breach process.
14. Suggestion to improve the paper: Perform regression analysis to develop mathematical models that relate the independent parameters to the peak outflow and erosion rates. This can help establish quantitative relationships and predict the impact of parameter changes on the dam breach process.
15. Suggestion to improve the paper: Conduct a sensitivity analysis to assess the sensitivity of the dam breach process to variations in the parameters. This involves systematically varying each parameter while keeping others constant and observing the resulting changes in the targets of the problem. Identify which parameters have the most significant impact on the dam breach process.
16. In the Conclusions section, it is important to highlight the following key points:
· Present key findings quantitatively and qualitatively and summarize their implications.
· Briefly recap the study's main objectives, methodology, novelty, and simulations.
· Highlight the significance of the results and their contribution to the existing body of knowledge.
· Address any limitations or potential areas for future research.
Comments on the Quality of English Language
The paper is full of grammatical and typographical errors (acomplex in Line 121; thesimulation in Line 203). The paper needs substantial editing to improve its quality. It is recommended to seek a professional editor's help to refine the paper.
Author Response
The authors thank the reviewer for the valuable and constructive comments on our manuscript, which we have found very helpful to the improvement of our paper. Please note that most of the changes were marked red in the revised manuscript. Our response to each comment is presented below.
- To enhance the introduction, I recommend incorporating new papers as well as the following ideas:
Background: Provide a brief overview of the significance of understanding and modeling the dam breaching process, emphasizing the importance of mitigating flood hazards, and minimizing their impact on people and structures.
Response: Thanks, a brief overview of the significance of understanding and modeling the dam breaching process, emphasizing the importance of mitigating flood hazards, and minimizing their impact on people and structures has been addressed in lines (42 to 46 and 49 to 55) and 2 new references have been added.
Summary of previous findings: Provide a concise summary of the results obtained from previous studies.
Response: Thanks, summary of the results obtained from previous studies is concluded in lines (113 to 116).
Research objective: Clearly state the objective of your study, which is to investigate the impact of different parameters on the dam breach process, specifically focusing on peak outflow and the rate of erosion.
Response: Thanks, objective of the study to investigate the impact of different parameters on the dam breach process, specifically focusing on peak outflow and the rate of erosion is stated and explained in lines (117 to 119).
Methodology: Mention the specific numerical modeling approach you used to conduct the study. Briefly explain why this methodology suits the research objective and review previous simulations.
Response: Thanks, all have been mentioned in lines (122 to 126).
Parameters under investigation: Highlight the parameters you examined in your study, including dam grain size materials, crest width, inflow discharge, and tailwater depth. Explain their relevance to the dam breach process and how they can influence peak outflow and erosion rates.
Response: Thanks, all have been highlighted in lines (119 to 122).
Implications and applications: Discuss the significance of the study findings for dam design and operation control. Emphasize how the results can be applied to improve dam stability and optimize the determination of crest width and tailwater depth.
Response: Thanks, This has been discussed and clarified in lines (125 to 127)
Importance: Conclude the introduction by reiterating the high importance of the study findings and how they contribute to the existing knowledge in dam breaching. Highlight the potential impact of this research on minimizing flood hazards and improving mitigation plans.
Response: Thanks, This point was concluded in lines (125 to 127).
- The first sentence of subsection 2.1 is repeated in the previous paragraph. Please delete it.
Response: Thanks, the repeated sentence has been deleted.
- Previous research shows several suitable software options for simulating the dam breaching process, including FLOW-3D HYDRO, HEC-RAS, MIKE 21, ICM, and HEC-HMS/HEC-FIA. What was the reason for using FLOW-3D HYDRO? Explain its strengths and advantages over other simulators.
Response: Thanks, The numerical models of the dam breach can be categorized according to flow dimensions (1D, 2D, or 3D). All these software’s are dependent on 1D or 2D for predicting outflow discharge. Generally, the accuracy of 2D models is still low, especially with velocity distribution over the flow depth, lateral momentum exchange, density-driven flows, and bottom friction [1]. Therefore, 3D models are preferred which is available in FLOW-3D HYDRO that can deal with such cases of dam breaching.
- Explain in full detail all assumptions and simplifications in the simulation process.
Response: Thanks, all assumptions and simplifications used in the simulation process have been added in lines (272 to 275)
- Why did you use the RNG k-e turbulence model? Have you evaluated the performance of other models, such as standard k-e and k-ω, through trial and error? If this is done, provide the results. Also, present the constants of the turbulence model in the relevant section.
Response: Thanks, in this study, we used RNG turbulence model because the majority of the previous studies indicate that the accuracy of RNG is high compared to other models specially, in sediment transport simulations. The RNG model uses equations like the equations for the k-e model. However, equation constants that are found empirically in the standard k-e model are derived explicitly in the RNG model. Generally, the RNG model has wider applicability than the standard k-e and k-ω models. In particular, the RNG model is known to describe low intensity turbulence flows and flows having strong shear regions more accurately [2]. All constants (C1, C2, C3) of the turbulence model were added in the relevant section from line161.
- The optimal grid significantly impacts the calculation cost and simulation accuracy. Present the results of the grid study in the form of a graph or the form of a table.
Response: Thanks, The graph and the reason of choice this size was added to the meshing of the model section lines (220 to 222)
- Show clearly the type of boundary condition in Figure 4. The current form is not legible.
Response: Thanks, the boundary conditions were defined clearly by adding coordinates directions for the model and defining boundaries according to it (lines 249 to 254)
- The paper is full of grammatical and typographical errors (acomplex in Line 121; thesimulation in Line 203). The paper needs substantial editing to improve its quality. It is recommended to seek a professional editor's help to refine the paper.
Response: Thanks, all the paper-language errors were adjusted.
- STL stands for what phrase? Mention the full form of abbreviations on first use.
Response: Thanks, The full form of abbreviation (STL) was mentioned in line 207.
- All equations must be referenced. Additionally, modify any incorrect subscripts and superscripts within the manuscript. Sort the nomenclature table alphabetically.
Response: Thanks, All equations references were added where the main reference is the FLOW-3D manual documents. The table of notation was rearranged alphabetically.
- What is the pressure-velocity coupling algorithm for the numerical solution? What techniques are used to discretize different equations? What are the convergence criteria for different equations? Add these items in full detail in the manuscript.
Response: Thanks, all items were added in full detail in the manuscript in lines (178 to 202).
- Simply presenting the results is inadequate; providing a thorough physical explanation for all observed alterations is essential. Expand the discussion on the results and organize a comparison with the results of other researchers to emphasize your findings. Since 2015, numerous papers close to this paper have emerged, offering opportunities for comparative analysis.
Response: Thanks, Explanation of findings was added in lines (316 to 319, 362 to 366, 395 to 397 and 422 to 424). Comparison of findings was added in lines (447 to 452).
- Suggestion to improve the paper: Explore the relationships between the parameters and the dam breach characteristics. Use correlation analysis to determine if there are any strong or weak correlations between the parameters and the peak outflow and erosion rates. This can help identify which parameters have the most influence on the dam breach process.
Response: Thanks, we agree with you that this could be an important point of research that we recommenced it future works and looking forward to doing it in a separate study.
- Suggestion to improve the paper: Perform regression analysis to develop mathematical models that relate the independent parameters to the peak outflow and erosion rates. This can help establish quantitative relationships and predict the impact of parameter changes on the dam breach process.
Response: Thanks, the current study focuses on numerical model but developing a mathematical model using regression analysis requires a massive work that we recommend for future work.
- Suggestion to improve the paper: Conduct a sensitivity analysis to assess the sensitivity of the dam breach process to variations in the parameters. This involves systematically varying each parameter while keeping others constant and observing the resulting changes in the targets of the problem. Identify which parameters have the most significant impact on the dam breach process.
Response: Thanks, we agree with you that this study improves the research quality, and we will study it in future works in this topic.
- In the Conclusions section, it is important to highlight the following key points:
Present key findings quantitatively and qualitatively and summarize their implications.
Response: Thanks, the results of all parameters and their implications were presented in summarized form in lines (461 to 470).
Briefly recap the study's main objectives, methodology, novelty, and simulations.
Response: Thanks, The main objectives were mentioned in lines (456 to 460). Methodology was included in lines (460 to 461). The novelty was presented in lines (469 to 470).
Highlight the significance of the results and their contribution to the existing body of knowledge.
Response: Thanks, The significance of the results was mentioned in lines (470 to 473).
Address any limitations or potential areas for future research.
Response: Thanks, The limitations for future research were presented in lines (473 to 478).
Reviewer 2 Report
Comments and Suggestions for Authors
This paper deals with a very interesting and important topic - the numerical investigation of the breach of embankment dam under different flow and material conditions. Still, there are some parts, which should be improved as follows from the following remarks and recommendations:
1. Some minor grammar corrections will be necessary. Some sentences should be rephrased.
2. Section 2.1: The theoretical part of the paper (except the turbulence modelling) refers just to the website of the software provider. Because the FLOW-3D code includes some unique features (the FAVOR concept, the TrueVOF technique with VOF particles, the sediment transport), more specific reference(s) should be used rather than the commercial webpage.
3. Section 2.4: The bottom of the coarser mesh block is defined as a volume flow rate boundary. How is the water inflow realized in the experiment? Authors should specify. Why the face between the coarser and finer mesh blocks is defined as a symmetry boundary? Isn't it just a general grid interface? The symmetry boundary condition acts as a mirror that reflects all the flow distribution to the other side. The conditions at symmetric boundary are no flow across boundary and no scalar flux across boundary. What are the initial values (including the water level height)?
4. Sections 3.1 and 3.2: What is the inflow discharge? Is it 1.0 lit/s? Authors should explicitly specify.
5. Sections 3.4: There is a significant effect of the tailwater depth on the erosion rate. Why? Authors should discuss in detail. Maybe, the evolution of longitudinal dam profiles with time could be useful and interesting.
6. Because the sides boundary conditions are defined as the walls, and at the dam crest, there is an initial breach, the 3D view of the dike as well of the water level during the timesteps would be highly recommended (at least for one study – the influence of inflow discharge or the influence of tailwater depth).
Comments on the Quality of English LanguageSome minor grammar corrections will be necessary. Some sentences should be rephrased.
Author Response
The authors thank the reviewer for their valuable and constructive comments on our manuscript, which we have found very helpful to the improvement of our paper. Please note that most of the changes were marked red in the revised manuscript. Our response to each comment is presented below.
- Some minor grammar corrections will be necessary. Some sentences should be rephrased.
Response: Thanks, the paper-language errors were adjusted.
- Section 2.1: The theoretical part of the paper (except the turbulence modelling) refers just to the website of the software provider. Because the FLOW-3D code includes some unique features (the FAVOR concept, the TrueVOF technique with VOF particles, the sediment transport), more specific reference(s) should be used rather than the commercial webpage.
Response: Thanks, other references were added in different parts such as Navier-Stokes equations and sediment transport model, but FLOW-3D is the main reference for the numerical model.
- Section 2.4: The bottom of the coarser mesh block is defined as a volume flow rate boundary. How is the water inflow realized in the experiment? Authors should specify. Why the face between the coarser and finer mesh blocks is defined as a symmetry boundary? Isn't it just a general grid interface? The symmetry boundary condition acts as a mirror that reflects all the flow distribution to the other side. The conditions at symmetric boundary are no flow across boundary and no scalar flux across boundary. What are the initial values (including the water level height)?.
Response: Thanks, the bottom of the coarser mesh block is defined as volume flow rate to become the same of the experimental work. In the experimental study, the bottom of flume was chosen to decrease the water surface Fluctuation.
Symmetry boundary condition is usually used to reduce computational effort during CFD simulation. This condition allows the flow to be transferred from one mesh block to another. No inputs are required for this boundary condition except that its location should be defined accurately. In coarser mesh block Xmax was symmetry and Xmin for finer mesh block, so that the flow entered to finer mesh block takes its data from coarser mesh by using this type of boundary.
The initial water level is just equal to the top level of the dam crest.
- Sections 3.1 and 3.2: What is the inflow discharge? Is it 1.0 lit/s? Authors should explicitly specify.
Response: Thanks, the inflow discharge value is 1.0 lit/s and mentioned in these sections in lines (300, 350 to 351).
- Sections 3.4: There is a significant effect of the tailwater depth on the erosion rate. Why? Authors should discuss in detail. Maybe, the evolution of longitudinal dam profiles with time could be useful and interesting.
Response: Thanks, the reasons which lead to decrease the erosion rate due to tailwater depth effect were presented in lines (423 to 425). A graph of dam profile evolution with time was also added (Figure 22).
- Because the sides boundary conditions are defined as the walls, and at the dam crest, there is an initial breach, the 3D view of the dike as well of the water level during the timesteps would be highly recommended (at least for one study – the influence of inflow discharge or the influence of tailwater depth).
Response: Thanks, the dam profile evolution with different timesteps was presented and the water surface is sheet on the dam profile (i.e., the surface of the water is following the dam profile formation) as shown in Figure 22.
References
[1] C.B. Vreugdenhil, Numerical methods for shallow-water flow, Springer Science & Business Media1994.
[2] M. Alemi, R. Maia, Numerical simulation of the flow and local scour process around single and complex bridge piers, International Journal of Civil Engineering 16(5) (2018) 475-487.
We would like to thank the reviewer for the valuable comments that have certainly improved the quality of the paper.
Round 2
Reviewer 1 Report
Comments and Suggestions for Authors
Accept